# LEARNING CHESS BLINDFOLDED

## ABSTRACT

Transformer language models have made tremendous strides in natural language understanding. However, the complexity of natural language makes it challenging to ascertain how accurately these models are tracking the world state underlying the text. Motivated by this issue, we consider the task of language modeling for the game of chess. Unlike natural language, chess notations describe a simple, constrained, and deterministic domain. Moreover, we observe that chess notation itself allows for directly probing the world state, without requiring any additional probing-related machinery. Additionally, we have access to a vast number of chess games coupled with the exact state at every move, allowing us to measure the impact of various ways of including grounding during language model training. Overall, we find that with enough training data, transformer language models can learn to track pieces and predict legal moves when trained solely from move sequences. However, in adverse circumstances (small training sets or prediction following long move histories), providing access to board state information during training can yield consistent improvements.

## 1 INTRODUCTION

Recently, transformer-based language models such as GPT-3 have stretched notions of what is possible with the simple self-supervised objective of language modeling, becoming a fixture in state of the art language technologies (Vaswani et al., 2017; Devlin et al., 2019; Brown et al., 2020). However, the black box nature of these models combined with the complexity of natural language makes it challenging to measure how accurately these models represent the world state underlying the text.

Motivated by the above issues, we propose training transformer language models for the game of chess. Chess provides a simple, constrained, and deterministic domain where the exact world state is known. Also, chess games can be transcribed exactly and unambiguously using chess notations (Section 2). In fact, the form of chess notations allows us to probe our language models for aspects of the board state using simple prompts (Section 3).

Due to the simplicity and precision of chess, we can evaluate language model predictions at a more fine-grained level than merely comparing it to the ground truth. For example, even if the next move prediction doesn't match the ground truth move, we can still evaluate whether the move is legal given the board state, and if it is illegal, we can determine the reason why (Appendix D). Moreover, since the state can be exactly modeled, we can evaluate models using counterfactual queries as well. The proposed evaluation sets and metrics are described in Section 5.3.

A side benefit of working with chess is that we have access to nearly unlimited data that is coupled with the exact board state at every turn. This board state is a form of grounding for the move sequence and allows us to compare training on move sequences alone to training with access to varying amounts of explicit state. Thus, modeling chess using language models may have implications for the debate surrounding the ability of language models to capture meaning if only exposed to text (Bender & Koller, 2020). To test the impact of chess board grounding on learnability and data efficiency, we can train language models with varying degree of access to the board state (Section 4).

Finally, while chess represents a controlled domain, it is by no means trivial for a language model. To illustrate the challenges of language modeling for chess, consider the left board shown in Figure 1b, where white is next to move. In order to generate a valid next move, the language model needs to (a) infer that it is white's turn, (b) represent the locations of all pieces, both white and black, (c) select one of the white pieces which can be legally moved, and finally (d) make a legal move with the selected

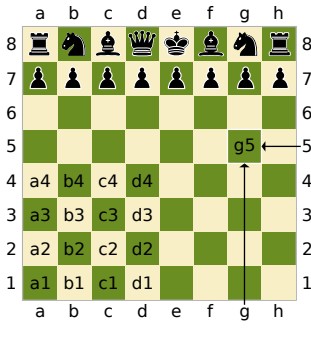
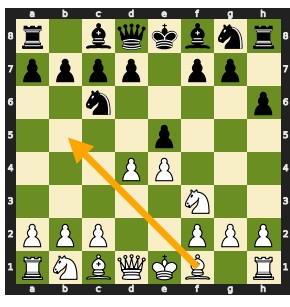
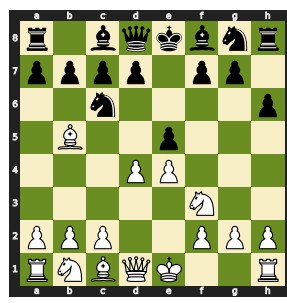

(a) Square naming

(b) Board state before (left) and after (right) the bishop at `f1` is moved to `b5`. UCI notation represents the move as `f1b5`.

Figure 1: Chess Notation

piece. Thus, a language model has to learn to track the board state, learn to generate moves according to the rules of chess, and on top of that learn chess strategies to predict the actual move.

We find that when given enough training data, transformers can learn to both track piece locations and predict legal moves at high levels of accuracy. However, when testing predictive ability for long move histories or when only given small training sets or when the model has access to limited history (Appendix C.1), predictive ability suffers. These challenging settings can provide an interesting testbed for future development of language models, and moreover because of the probing properties, errors can be diagnosed in great detail. In these more challenging settings, we show that providing parts of the board state (during training time only) can lead to significant improvements in accuracy.

Our results also provide some key insights on transformer language models: (i) They are robust to various ways of incorporating explicit supervision about the board state when given enough training data. (ii) In particular, they are robust to changes in input distribution where additional tokens, related to board state, are added to input sequence *only during training* (Section 4.1). In contrast to LSTMs, transformers achieve this robustness even with smaller training sets (Appendix F). (iii) The model performance strongly relies on access to the whole sequence history as the performance drops on limiting this access to a fixed-size window of previous tokens (Appendix C.1).

To summarize, our contributions are to:

- Propose chess as a testbed for evaluating world state tracking capabilities of language models.
- Show that by selecting (and tweaking) the appropriate chess notation, we can probe language model for aspects of the world state using simple prompts (Section 3).
- Propose a suite of probing tasks to evaluate language models for chess on world state tracking (Section 5.3). These probing tasks go beyond simple exact match, and use a more fine-grained evaluation, and allow for automated error analysis (Appendix D).
- Show that given enough training data, transformer language models can learn to track piece locations and predict legal moves at high levels of accuracy.
- Evaluate the effect of grounding by training and evaluating a spectrum of transformer language models with varying degrees of access to the world state. We find that grounding helps in challenging settings of our proposed probing tasks.
- Provide insights on transformer language models such as their robustness to incorporating the world state in various ways, their dependence on access to the whole history, etc.

## 2 BACKGROUND

**Chess Preliminaries.** Figure 1a[1] shows how squares are indicated in chess notations via a combination of lettered columns and numbered rows. Chess notations use this square naming convention to denote the movement of pieces. As our notation, we choose Universal Chess Interface (UCI) notation, which combines the starting square and the destination square to represent a move.[2] The move in Figure 1b is

---

[1]Source `https://en.wikipedia.org/wiki/File:SCD_algebraic_notation.svg`
[2]For more details see `https://en.wikipedia.org/wiki/Universal_Chess_Interface`

represented as `f1b5` in UCI where `f1` indicates the starting square and `b5` denotes the ending square. While another notation, SAN, is the standard choice for gameplay, we prefer UCI (see Appendix A for our reasons for choosing UCI over SAN).

## 2.1 Related Work

**Simulated Worlds and Grounding.** There have been several prior efforts in relating simulated worlds to natural language. The bAbI framework simulates a world modeled via templates to generate question-answering (QA) tasks (Weston et al., 2015a). The recent TextWorld framework facilitates generating, training, and evaluating interactive text-based games (Côté et al., 2018). bAbI and TextWorld are similar to our work in the sense that the true world state is, by construction, available. The key difference with bAbI is that the it provides explicit world-state supervision in the form of training data for QA. With TextWorld we differ in: (1) their objective (reward maximization vs. maximizing the probability of the next observation), (2) how they are ultimately evaluated (final reward vs. world state tracking), and (3) whether we can directly probe the model's knowledge of the entire state. The world models of Ha & Schmidhuber (2018) also maximize the probability of the next observation, but differ along the other two dimensions. Similarly the work by Hermann et al. (2017) and Hill et al. (2017) on developing and using 3D world simulations for learning grounded language has only partial overlap with the objective function, and differ along the other two aspects.

Our work is related to work on grounding in that we are interested in comparing model performance when it does not have access to grounding information to when it does (Bruni et al., 2014; Kiros et al., 2014; Ororbia et al., 2019). However, unlike the work of Ororbia et al. (2019), for instance, the goal is not to improve performance of language models using access to more of the world state, but to assess how much of this state has been recovered by the model from just learning the LM task.

**Cloze Tasks for Natural Language Models.** There has been a plethora of prior work on developing and using cloze tasks for evaluating natural language models (Hermann et al., 2015; Hill et al., 2016). These cloze tasks can range from testing general text understanding (Paperno et al., 2016) to targeting particular aspects of natural language, such as commonsense/pragmatics (Mostafazadeh et al., 2016; Ettinger, 2020), narrative understanding (Mostafazadeh et al., 2017), factual knowledge (Petroni et al., 2019), etc. Creating these tasks often requires human curation,[3] and the evaluation is typically limited to exact match. In contrast, we propose cloze tasks/prompts targeting the world state, which can be precisely automated for chess, and can be evaluated at a fine-grained level.

**Probing.** One of the goals of this work is to probe the language model's board state tracking capability. A typical solution used by prior work is to train a probing model on top of a pretrained model (Ettinger et al., 2016; Alain & Bengio, 2017; Adi et al., 2017; Tenney et al., 2019; Hewitt & Liang, 2019). This setup is time-consuming as it requires training probing models for all tasks. Moreover, the complexity of the probing model can also affect the conclusions (Pimentel et al., 2020). In our case, we show that by appropriate choice of notation, probing for board state can be accomplished via simple prompts (Section 3).

**Deep Learning for Chess.** Deep networks have been used in prior work to predict/mimic the next move given the true game state David et al. (2016); Oshri & Khandwala (2015). Using just self-play and the rules of chess, AlphaZero achieves superhuman performance starting from random play (Silver et al., 2018). The focus of these work is the quality of game play given the true board state while we just use chess as a testbed for evaluating the tranformer LMs world state tracking capabilities. Recently there have been several work focusing on transformer language models for chess (Presser & Branwen, 2020; Cheng, 2020; Noever et al., 2020). These work are similar to ours in the sense that input is limited to the move sequence and not the true board state, but the focus of these work is again the quality of game play rather than how well is the model aware of the underlying board state.

---

[3] Automated cloze tasks without any human filtering can end up with instances which even humans can't answer (Hill et al., 2016)

Table 1: Token sequences corresponding to the move sequence `e2e4 e7e5 g1f3` for different notations during training and inference. Notice that regardless of the RAP probability used during training, at inference time the token sequences have no piece types.

| Notation | Training | Inference |
|---|---|---|
| UCI | e2, e4, e7, e5, g1, f3 | e2, e4, e7, e5, g1, f3 |
| UCI + RAP 15 | e2, e4, P, e7, e5, g1, f3 | e2, e4, e7, e5, g1, f3 |
| UCI + RAP 100 | P, e2, e4, P, e7, e5, N, g1, f3 | e2, e4, e7, e5, g1, f3 |

# 3    LANGUAGE MODEL PROMPTS AS BOARD STATE PROBES

Chess notations provide a simple alternate solution of using language model prompts as board state probes. For example, the prompt "e2e4 e7e5 g1f3 b8c6 d2d4 h7h6 f1" (the underlined move sequence leads to the left board state in Figure 1b) can be used for next-token prediction with a language model trained on UCI notation. The generated token can be interpreted as the ending square predicted for the bishop at `f1`. This prediction can then be used to determine the level of board state awareness of the model. A prediction of `g1` may indicate that the model does not recognize that the piece type at `f1` is a bishop, as such a move is not possible for a bishop. If the model predicts `g2`, it may mean that the model is not aware that another piece is currently located at `g2`.

UCI notation is sufficient for assessing ending square prediction. However, it does not allow us to test starting square prediction directly. That is, we would like to give a language model the prompt "e2e4 e7e5 g1f3 b8c6 d2d4 h7h6 N", where N represents knight, and expect it to generate a valid starting position for a knight of the correct color. To allow testing the model with such prompts, we propose randomly including piece types in moves during training with some fixed probability $p$. We refer to this strategy as "randomly annotated piece type" (RAP) and use the nomenclature "UCI + RAP $p$" to indicate that with $p$% probability, piece type is part of the move notation during training. Note that for $p = 0$, the notation reduces to UCI. When *testing* with these starting square prediction prompts, we only include piece type for the prompt, not for any moves in the history. Thus, using RAP during training allows us to probe, at test time, where the model thinks each piece is, given any game history's prefix; by simply providing the desired piece type (e.g., N) the model outputs the predicted starting square for a piece of that type.

# 4    LANGUAGE MODELS AND VARIANTS

We use the GPT2-small architecture for our base language model (Vaswani et al., 2017; Radford et al., 2019). GPT2-small is a 12-layer transformer model with 12 attention heads and an embedding size of 768 dimensions. The context size of the model is limited to 512, which is sufficient to cover the longest game in our training set. Note that we only borrow the model architecture; the models themselves are trained from scratch. We use a simple regular expression based tokenizer, which considers a board square symbol such as `b1` as a single token. This gives us a vocabulary of 77 tokens which includes the 64 squares, piece type symbols, and some other special symbols (see Table 5 in the Appendix).[4]

We next discuss variants of the base language model that incorporate the board state in various ways.

## 4.1    RANDOMLY ANNOTATED PIECE TYPE (RAP)

One way of introducing an aspect of board state into move sequences is by randomly annotating piece types (RAP) in training sequences, as introduced in Section 3. Table 1 illustrates how the use of RAP changes the token sequence during training but not during inference. We hypothesize that adding piece types during training can aid the model in learning to track the board state by providing additional supervision. Similar ideas have been used when training memory networks for tasks involving entity tracking in natural language (Weston et al., 2015b; Hill et al., 2016).

---

[4]In initial experiments we used a delimiter token to indicate move boundary. However, removing it did not degrade performance, and made training much faster due to reduced sequence length.

## 4.2 INCORPORATING BOARD STATE

One of the criticisms sometimes leveled against language models for natural language is that they can never understand the "meaning" of language, particularly given that they do not use any extra-linguistic information, such as visual grounding (Bender & Koller, 2020). The simple and deterministic dynamics of chess means that chess offers nearly-unlimited data with perfect grounding. The ground truth board state can be used to learn a grounded language model, and can also be used in oracle settings which use this board state during both training and inference. We next describe details of the board state representation followed by its incorporation in the base language model.

### 4.2.1 BOARD STATE REPRESENTATION

Following Oshri & Khandwala (2015), we represent a chess board state as an $8 \times 8$ image with six channels (see Figure 3 in the Appendix). The six channels correspond to the six unique piece types in chess, namely, pawn, rook, bishop, queen, king, and knight. The channel for a given piece type has a value of +1 at locations of its white instances, and -1 at locations of black instances, with 0 indicating empty positions. Apart from the piece positions, we add a seventh channel to represent additional information about the board state. In particular, we use one bit to indicate which player moves next, one bit to indicate if the current player's king is in check, and another four bits to represent castling rights of the two players.[5][6] These six bits are represented as another channel by padding with zeros to make it an $8 \times 8$ grid. Thus, we have a $7 \times 8 \times 8$ tensor for the raw board representation.

The raw board representation is fed to a 2-layer convolutional neural network (CNN). Both CNN layers have 128 filters of kernel size $2 \times 2$ with ReLU activations. The output of the CNN is passed through a linear layer which outputs a final 768-dimensional board state representation.

### 4.2.2 MULTI-VIEW TRAINING

We use the board state representation during training to learn a grounded language model. Let $(\boldsymbol{h}_t^i)_{i=0}^{12}$ represent the transformer language model's hidden states at timestep $t$, where the $\boldsymbol{h}_t^0$ are simply the embeddings for the input symbols plus the position embeddings. Then, the language model's state representation $\boldsymbol{l}_t$ is a learned convex combination of the transformer hidden states:

$$\boldsymbol{l}_t = \sum_{k=0}^{12} \alpha_k \boldsymbol{h}_t^k \qquad \boldsymbol{\alpha} = \mathrm{softmax}(w_0, w_1, \cdots, w_{12})$$

where $\boldsymbol{w} \in \mathbb{R}^{13}$ is a learned parameter vector.

Given a minibatch of tokenized chess games, let $K$ be the total number of moves in this minibatch. Let $(\boldsymbol{l}_i)_{i=0}^{K}$ represent the language model state representations at the end of each move in the minibatch, and let $(\boldsymbol{s}_i)_{i=0}^{K}$ represent the corresponding board state representations. We want the two representations to be close to each other when they correspond to the same state, and far apart otherwise. To promote this behavior, we use a "quadruplet" loss (Chen et al., 2017):

$$\mathcal{L}(\boldsymbol{l}_i, \boldsymbol{s}_i) = \max(0, m + d(\boldsymbol{l}_i, \boldsymbol{s}_i) - d(\boldsymbol{l}_i, \boldsymbol{s}_i')) + \max(0, m + d(\boldsymbol{l}_i, \boldsymbol{s}_i) - d(\boldsymbol{l}_i', \boldsymbol{s}_i))$$

where $m$ represents the margin by which we want the positive pairs to be closer than the negative pairs, $d(., .)$ represents the distance function (in our case, cosine distance), $\boldsymbol{l}_i'$ is a language model state which corresponds to a board state other than $\boldsymbol{s}_i$, and $\boldsymbol{s}_i'$ is a board state which corresponds to a language model state other than $\boldsymbol{l}_i$. For sampling a negative state given the other view's state, we randomly draw 10 states from the minibatch and pick the one closest to the other view's state. For margin, we use $m = 0.6$ as it gets the lowest validation perplexity among $\{0.2, 0.4, 0.6, 0.8\}$. Finally, the quadruplet loss is simply added to the language modeling loss without any scaling.

### 4.2.3 ORACLE BASELINE

Just for comparative purposes, for our final baseline  we assume that the language model has access to both the piece type with which moves are made (e.g., `Bf1b5` instead of `f1b5`) and the board state

---

[5]Details on castling rights: `https://www.chessprogramming.org/Castling_Rights`
[6]The board state used is incomplete. We skip details such as *en passant*, *threefold repetition*, *fifty-move rule counter*, etc. as they come into play only in rare cases.

Table 2: Examples of each probing task, as well as the corresponding exact move (ExM) and legal move (LgM) correct answers, are shown below. All examples assume the language model was fed the prefix `e2e4 e7e5 g1f3 b8c6 d2d4 h7h6` (see Figure 1b), and that the actual next move was `f1b5`. While there is only one valid prompt token for both End-Actual and Start-Actual tasks, there are many valid prompt tokens for the other tasks, and we show just one possibility for each. Start-tasks (bottom sub-table) assume the model was trained on games described in UCI+RAP notation.

| Task | Prompt Token | Correct Answers (ExM) | Correct Answers (LgM) |
|------|------|------|------|
| End-Actual | `f1` | {`b5`} | {`e2`, `d3`, `c4`, `b5`, `a6`} |
| End-Other | `f3` | N/A | {`d2`, `g1`, `h4`, `g5`, `e5`} |
| Start-Actual | `B` | {`f1`} | {`f1`, `c1`} |
| Start-Other | `N` | N/A | {`f3`, `b1`} |

at the end of every move during both training and inference. Access to piecetype can limit the set of moves to consider, and the access to board state can relieve the language model from the burden of board state tracking. To incorporate the board state, we simply sum the board state with the hidden state of all the layers of the transformer at the end of each move.[7] This baseline serves as an *approximate* upper bound to all other model variants described above.

## 5 EXPERIMENTAL SETUP

### 5.1 DATA

We use the Millionbase dataset which is freely available and has close to 2.9 million quality chess games.[8] After filtering out duplicate games, games with fewer than 10 moves, and games with more than 150 moves (for the complete game to fit into one transformer window), we are left with around 2.5 million games. From this filtered set we randomly select 100K games for training, 15K games each for dev and test, and another 50K games to create prompt-based evaluation sets described in Section 5.3. The dev and test set are used for calculating perplexities on them. The dev set perplexity is used for choosing the hyperparameters for models which incorporate the board state. From the 100K training set, we create subsets of size 15K and 30K which we refer to as "Train-S" and "Train-M", while the full training set is referred to as "Train-L". For detailed statistics, see Table 9 in Appendix. All the data processing steps requiring chess knowledge, including parsing chess databases, are carried out using python-chess (Fiekas, 2012).

### 5.2 TRAINING DETAILS

Models are trained for 10 epochs with a batch size of 60. Validation is performed at the end of every epoch and training is halted whenever the validation loss starts increasing. For optimization we use Adam (Kingma & Ba, 2014) with learning rate of $5 \times 10^{-4}$ and L2 weight decay of 0.01. The learning rate is warmed up linearly over the first 10% of training followed by a linear decay. To accelerate training, we use mixed precision training (Micikevicius et al., 2018). All experiments are carried out using the PyTorch Lightning framework which is built on top of PyTorch (Falcon, 2019; Paszke et al., 2019). We use the GPT-2 implementation from the transformers library (Wolf et al., 2019).

For the UCI + RAP $p$ models, we try $p \in \{0.01, 0.05, 0.15, 0.85, 1.00\}$, and find $p = 0.05$ and $p = 0.15$ get the best perplexity on the dev set. Larger values of $p$ may lead to greater mismatch between training and inference, while small values like $p = 0.01$ likely do not provide enough training signal.

### 5.3 BOARD STATE PROBING TASKS

In this section we describe how we probe the language model's board state understanding. In each probing task we feed the model a prefix of a game followed by a single prompt token, and the model is

---

[7]Other strategies for fusing the two views (Ororbia et al., 2019; Ziegler et al., 2019) are left for future work.

[8]Download link available at `https://rebel13.nl/rebel13/rebel%2013.html`

evaluated based on the highest probability next-token under the model given this context. We show an example of each probing task in Table 2 (which we further describe below), assuming the model has been fed the move sequence prefix `e2e4 e7e5 g1f3 b8c6 d2d4 h7h6`, which is visualized as the left board in Figure 1b. The actual next move played in the game is `f1b5`, which takes the white bishop at square `f1` to square `b5`, as shown in the right board of Figure 1b.

To create these evaluation sets, we use the 50K unique games reserved for this task (Section 5.1). We only consider prompts for non-pawn pieces since the dynamics of pawns are fairly limited. We ensure that the game prefixes selected are never seen in the training data. To see the effect of game prefix length $l$ (measured in number of moves) on these predictions, we create two classes of this task: (i) Short, where $5 \leq l \leq 50$, and (ii) Long, where $51 \leq l \leq 100$. For both classes we create evaluation sets with 1000 instances. Next we describe the various probing tasks in detail.

### 5.3.1 ENDING SQUARE TASKS

In this set of tasks, the model is given a game prefix and prompted with the starting square of the next move (`f1` in the example of Table 2). The model's next-token prediction represents its prediction for the ending square of this move, which tests the model's ability to track the board state and follow the rules of chess, as well as strategic awareness.[9] We consider two task variants:

1. **End-Actual**: Given a move sequence prefix, the model is prompted with the starting square of the actual piece moved next in the game.
2. **End-Other**: Given a move sequence prefix, the model is prompted with the starting square of any piece on the board that can be legally moved according to rules of chess.

We evaluate End-Actual predictions in terms of both exact move (ExM) accuracy (whether the model predicted the true ending square, `b5` in our running example) and legal move (LgM) accuracy (whether the model predicted a legal ending square for the piece starting at the square in the prompt). ExM accuracy evaluation is similar to the typical evaluation of language models on natural language data, while LgM is less stringent and focuses on testing just the model's understanding of chess rules and the board state. Note that for End-Other, only LgM accuracy is available. See Table 2 for examples.

### 5.3.2 STARTING SQUARE TASKS

In this category of task, the model is again given a game prefix, but prompted with just the piece *type* of the next move, such as `B` for bishop in the example in Table 2. The model's next-token prediction thus represents its prediction for where the prompted piece type currently is on the board. This task tests the model's ability to track pieces.[10] Note that only models which have seen piece types during training, i.e. "UCI + RAP" models, can actually be tested on this task. Also, no piece types are used in the game prefix (except for the oracle baseline). We again have two variants of this task:

1. **Start-Actual**: Given a move sequence prefix, the model is prompted with the piece type of the actual piece moved next in the game.
2. **Start-Other**: Given a move sequence prefix, the model is prompted with the piece type of any piece on the board that can be legally moved according to rules of chess.

We again evaluate Start-Actual predictions both in terms of ExM accuracy (whether the model predicts the starting square of the piece actually moved next in the game), as well as in terms of LgM accuracy (whether the model predicts the starting square of a piece with the correct piece type). For Start-Other, only LgM accuracy is available; see Table 2 for examples.

## 6 RESULTS

To save space, perplexity results are presented in Table 10 of Appendix. Table 3 shows results when predicting starting squares and Table 4 shows results when predicting ending squares. We will divide our discussion of the results into those that pertain to board state tracking and those that pertain to modeling chess strategy.

---

[9]Strategic capabilities of a chess language model are strongly tied to the quality of training games.

[10]In certain cases, this task also tests understanding of chess rules. For example, in Figure 1b only the rook at `h1` can be moved.

Table 3: Accuracies (%) for predicting starting squares ("Start-Actual" and "Start-Other" tasks)

| Training Set | Model | Starting Square (Short) | | | Starting Square (Long) | | |
| | | Actual | | Other | Actual | | Other |
| | | ExM | LgM | LgM | ExM | LgM | LgM |
|---|---|---|---|---|---|---|---|
| | UCI + RAP 5 | 71.3 | 88.8 | 83.2 | 58.6 | 69.5 | 65.4 |
| Train-S | UCI + RAP 15 | 80.4 | 96.7 | 93.8 | 72.6 | 81.6 | 79.4 |
| | Oracle Baseline | 86.3 | 99.1 | 97.2 | 89.3 | 99.0 | 98.5 |
| | UCI + RAP 5 | 78.8 | 95.1 | 92.4 | 70.9 | 80.9 | 78.4 |
| Train-M | UCI + RAP 15 | 84.5 | 98.7 | 97.1 | 83.0 | 93.3 | 92.3 |
| | Oracle Baseline | 87.1 | 99.7 | 98.1 | 90.2 | 99.6 | 99.4 |
| | UCI + RAP 5 | 86.8 | 99.7 | 97.9 | 88.6 | 98.0 | 98.3 |
| Train-L | UCI + RAP 15 | 87.4 | 99.7 | 98.2 | 89.6 | 99.3 | 98.8 |
| | Oracle Baseline | 90.2 | 99.9 | 98.9 | 91.7 | 99.6 | 99.5 |

## 6.1 BOARD STATE TRACKING

There are several observations to note. First, **transformers can learn to identify where pieces are located**. This is shown by the "LgM" accuracies in Table 3. UCI + RAP 15 can predict a valid starting position of a piece at 99.7% accuracy for short histories and 99.3% accuracy for long histories. When asked to identify the location of a piece other than the one selected to be moved next, these accuracies drop only slightly, to 98.2% and 98.8%, respectively. However, this capability requires the large training set. For the medium training set and long histories, accuracy drops to 93.3%, and with the small training set, it drops further to 81.6%.

RAP 5 reaches the same starting position accuracy as RAP 15 for short histories, but for long histories there is a 1.3% gap. At smaller training sizes, the superiority of RAP 15 becomes much clearer, showing the benefit of providing piece type information more frequently during training.

The difference between the location of the piece in the exact move (ExM) and the location of either piece of the given type (LgM) is substantial, at more than 10% absolute. However, this difference relates to chess strategy rather than board state tracking. When querying a piece type other than the one moved, the accuracies are still high for RAP 15, at 98.2% for short histories and 98.8% for longer histories, though slightly lower than when querying the actual piece type. This trend suggests that piece location tracking is slightly better for the piece type that is actually moved.

Second, **transformers can learn to predict legal moves**. This is shown by the "LgM" accuracies in Table 4, for which UCI + RAP 15 exceeds 99% for short histories. We do observe a drop to 95.1% for long histories, which is likely because board state tracking becomes more difficult with longer histories. When decreasing training data, the short history LgM accuracy drops by 2% in each case, but the long history accuracy drops by 5%. It may be the case that training on an even larger training set can improve long-history accuracy further.

Overall, when enough training data is available, randomly adding piece types (with RAP 15) is similar to using the complete board state via multi-view training. With small training sets, the multi-view model struggles to use board states effectively, but with the large training set, the results are slightly better than RAP 15. Transformer language models are robust to various ways of incorporating explicit supervision about the board state when given enough training data.

We find consistent gains in accuracy when randomly adding piece types during training. UCI + RAP 5 improves over UCI, which corresponds to RAP 0, and UCI + RAP 15 improves further in most settings, with the largest improvements for Train-S. With Train-L, UCI is able to predict a legal ending square in 98.6% of cases when using short histories. This strong result indicates that a transformer without any form of board state supervision can still learn to track board state well enough to predict ending squares if given enough data. A detailed error analysis of illegal moves for the Train-L setting is presented in Appendix D.

Given the strong results with large training sets, future work can focus on challenging settings such as: (a) low data regime, (b) long(er) game histories, (c) limited access to previous tokens in self-attention (Appendix C.1), (d) predicting more board state variables simultaneously, etc.

Table 4: Accuracies (%) for predicting ending squares ("End-Actual" and "End-Other" tasks)

| Training Set | Model | Ending Square (Short) | | | Ending Square (Long) | | |
| | | Actual | | Other | Actual | | Other |
| | | ExM | LgM | LgM | ExM | LgM | LgM |
|---|---|---|---|---|---|---|---|
| Train-S | UCI | 51.9 | 90.8 | 84.3 | 28.3 | 69.5 | 64.8 |
| | UCI + RAP 5 | 53.4 | 94.4 | 86.5 | 32.5 | 78.7 | 73.6 |
| | UCI + RAP 15 | 57.4 | 94.8 | 89.2 | 36.1 | 84.6 | 76.4 |
| | UCI + Multi-view Training | 51.4 | 90.2 | 85.1 | 27.7 | 71.0 | 65.8 |
| | Oracle Baseline | 59.0 | 95.7 | 89.9 | 39.4 | 89.4 | 84.3 |
| Train-M | UCI | 56.8 | 95.1 | 90.7 | 35.2 | 82.8 | 76.9 |
| | UCI + RAP 5 | 59.8 | 97.0 | 91.0 | 39.6 | 88.9 | 81.3 |
| | UCI + RAP 15 | 61.1 | 96.9 | 90.5 | 43.0 | 90.2 | 83.7 |
| | UCI + Multi-view Training | 56.8 | 95.9 | 91.7 | 35.7 | 83.3 | 78.6 |
| | Oracle Baseline | 62.1 | 97.8 | 93.1 | 46.0 | 92.4 | 87.4 |
| Train-L | UCI | 66.7 | 98.6 | 94.9 | 47.1 | 94.5 | 89.9 |
| | UCI + RAP 5 | 69.0 | 98.9 | 94.8 | 50.3 | 95.6 | 91.7 |
| | UCI + RAP 15 | 68.6 | 99.1 | 95.2 | 48.4 | 95.1 | 91.6 |
| | UCI + Multi-view Training | 69.7 | 99.2 | 95.8 | 49.3 | 95.9 | 92.3 |
| | Oracle Baseline | 69.1 | 99.1 | 95.9 | 53.8 | 97.4 | 94.0 |
| | Random Legal Move | 26.2 | - | - | 21.8 | - | - |

## 6.2 CHESS STRATEGY MODELING

Overall, our results show that predicting the actual moves made by human players is much more difficult than identifying valid piece positions and legal moves.

We can view the exact move accuracy in starting square prediction (Table 3) as the ability to determine which piece to move next given the piece type. The Oracle Baseline ExM results show the difficulty of this task even when the full board state is provided at inference time. With the large training set, we see gaps of 2-4% between the RAP models and the Oracle Baseline, while in terms of LgM, which does not involve strategy, the gaps are much smaller. In general, the gaps increase with less training data and longer histories.

When predicting ending squares (Table 4), the relatively low ExM accuracies of the Oracle Baseline show the difficulty of the task. The strongest models without board state during inference—-RAP 5, RAP 15, and multi-view—-are able to match or exceed the Oracle Baseline when having access to the large training set and for short histories. This result shows that our models are not limited by their ability to track the board state over short histories, but rather by the inherent difficulty in predicting the next move performed by a human player. However, for long histories, the Oracle Baseline is consistently more accurate than our models.

## 7 CONCLUSION

We propose the task of language modeling for chess to evaluate how well language models can capture the world state. We show that with appropriate choice of chess notation, a language model can be probed for different aspects of the board state via simple prompts. The simple and precise dynamics of chess allow for (a) training models with varying amount of explicit state, and (b) evaluating model predictions at a fine-grained level. Results indicate that transformer language models are able to track the board state when given enough data, but in adverse circumstances, providing access to board state information during training can yield consistent improvement.

**Wider Implications for Natural Language Processing.** Our results shed light on the following interesting properties of transformers: (a) they are robust to RAP-like changes in input distribution, and (b) they require access to long context (Appendix C.1), and large training sets. Future work can use the first finding to introduce the world state, or more specifically the output of linguistic analyzers such as coreference, via RAP-like tokens during pre-training and fine-tuning of transformers. RAP-like tokens can also be used for debugging/diagnosing the model's understanding, similar to the starting square prediction tasks. The second finding can be another motivation to search for new architectures that are adept at understanding long text with access to limited history (Rae et al., 2020), and that require small training sets. The proposed framework allows for probing and understanding new architectures that address these challenges.

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

## A  SAN Notation

**Standard Algebraic Notation (SAN)**    combines the piece type moved and the destination square to denote a move.[11] For example, the move in Figure 1b is represented as `Bb5` in SAN where `B` represents the piece type bishop and `b5` represents the destination square.

**Standard Algebraic Notation (SAN) Ambiguity**    SAN notation doesn't use the starting square of the piece in its move representation. This limits the ability to prompt a SAN-based language model with specific piece type instances. For example, given the prompt "`e4 e5 Nf3 Nc6 d4 h6 B`" (the underlined move sequence leads to the left board state in Figure 1b), it's not clear whether the token `B` refers to the bishop at `f1` or `c1`. Due to this limitation on the specificity of probing queries, we do not use SAN for our experiments.

## B  Model Vocabulary

Table 5: Model Vocabulary

| Type | Examples | Count |
|------|----------|-------|
| Square names | e4, d1 | 64 |
| Piece type | P, K, Q, R, B, N | 6 |
| Promoted Pawn Piece type | q, r, b, n | 4 |
| Special symbols | BOS, EOS, PAD | 3 |
| Total | | 77 |

Table 5 shows the vocabulary used in our experiments. We don't use any delimiter token to denote the move boundary. Tokens of promoted pawn piece type are used when a pawn gets promoted. For example, `e7e8q` denotes the move where a pawn from e7 moves to e8 and becomes a queen.

## C  Other Model Configurations

In this section, we explore the effect of varying the transformer architecture along two dimensions: (a) limiting the attention window, and (b) increasing the model size. These experiments are designed to help ascertain what challenges remain in tracking the state of a chess game with a language model.

### C.1  Limited History

Chess is Markovian, and the exact board state can be represented in less than 1000 bits.[12] Thus, the high-dimensional multi-layered transformer state representation can in theory represent the board state in the hidden state of even a single timestep. However, the transformer LM has no chess domain knowledge, and thus will likely learn a very different state representation. In this ablation, we want to inspect how much is any particular time step's representation important to representing the game's state. Specifically, is it important for the model to attend to all the previous tokens or the model can learn a compressed state representation if forced to do so.   To this end, we consider the case where the model can only attend to the current token and the previous $w$ tokens. Note that the baseline model has access to the entire history.

Table 6 presents results comparing the baseline model with models which have access to only the past 10 or 50 hidden states. In comparison to the baseline, UCI ($w = 10$) suffers a significant drop in almost all evaluations except for accuracy over short histories with the Train-L training set. UCI ($w = 50$) fares better in comparison to UCI ($w = 10$) and matches the baseline on LgM accuracies for short histories given enough training data. However, the same is not true for evaluations over long histories, where the gap between UCI and UCI ($w = 50$) seems to widen with an increase in training

---

[11]For more details see `https://en.wikipedia.org/wiki/Algebraic_notation_(chess)`
[12]`https://en.wikipedia.org/wiki/Forsyth%E2%80%93Edwards_Notation`

Table 6: Accuracies (%) for predicting ending squares ("End-Actual" and "End-Other" tasks) with varying attention window sizes.

| Training Set | Model | Ending Square (Short) | | | Ending Square (Long) | | |
| | | Actual | | Other | Actual | | Other |
| | | ExM | LgM | LgM | ExM | LgM | LgM |
|---|---|---|---|---|---|---|---|
| | UCI | 51.9 | 90.8 | 84.3 | 28.3 | 69.5 | 64.8 |
| Train-S | UCI ($w = 50$) | 49.0 | 88.5 | 81.7 | 25.7 | 68.1 | 61.7 |
| | UCI ($w = 10$) | 43.5 | 79.7 | 69.9 | 22.4 | 62.9 | 50.6 |
| | UCI | 56.8 | 95.1 | 90.7 | 35.2 | 82.8 | 76.9 |
| Train-M | UCI ($w = 50$) | 53.6 | 94.6 | 88.6 | 32.9 | 79.7 | 74.4 |
| | UCI ($w = 10$) | 50.9 | 86.1 | 77.3 | 27.7 | 68.9 | 57.6 |
| | UCI | 66.7 | 98.6 | 94.9 | 47.1 | 94.5 | 89.9 |
| Train-L | UCI ($w = 50$) | 64.4 | 98.7 | 94.4 | 42.1 | 89.9 | 85.7 |
| | UCI ($w = 10$) | 60.0 | 95.5 | 87.9 | 36.5 | 81.0 | 73.1 |

data. These results clearly demonstrate the dependence of the transformer LM's performance on access to unrestricted history, which is qualitatively different from an ideal state representation for chess.

## C.2 LARGER MODELS

Table 7: Accuracies (%) for predicting ending squares for different model sizes. GPT2-small = {12 layers, 12 heads, 768 embedding size}; GPT2-intermediate = {16 layers, 12 heads, 768 embedding size}; and GPT2-medium = {24 layers, 16 heads, 1024 embedding size}.

| Training Set | Model | Ending Square (Short) | | | Ending Square (Long) | | |
| | | Actual | | Other | Actual | | Other |
| | | ExM | LgM | LgM | ExM | LgM | LgM |
|---|---|---|---|---|---|---|---|
| | GPT2-small | 51.9 | 90.8 | 84.3 | 28.3 | 69.5 | 64.8 |
| Train-S | GPT2-intermediate | 50.8 | 90.0 | 85.3 | 26.4 | 68.2 | 65.3 |
| | GPT2-medium | 50.6 | 88.6 | 83.9 | 24.1 | 65.8 | 60.7 |
| | GPT2-small | 56.8 | 95.1 | 90.7 | 35.2 | 82.8 | 76.9 |
| Train-M | GPT2-intermediate | 60.2 | 96.1 | 90.6 | 35.1 | 82.5 | 77.3 |
| | GPT2-medium | 57.7 | 94.6 | 89.4 | 34.3 | 79.7 | 73.5 |
| | GPT2-small | 66.7 | 98.6 | 94.9 | 47.1 | 94.5 | 89.9 |
| Train-L | GPT2-intermediate | 67.5 | 98.8 | 95.2 | 48.3 | 95.8 | 91.7 |
| | GPT2-medium | 67.0 | 98.6 | 94.7 | 49.0 | 95.6 | 91.0 |

Table 7 presents results with transformer models of sizes varying from GPT2-small to GPT2-medium. We also introduce a new configuration, referred to as GPT2-intermediate, which serves as an intermediate between GPT2-small and GPT2-medium. For Train-S, GPT2-small outperforms GPT2-medium on all evaluations, and outperforms GPT2-intermediate on all but LgM accuracies for the "Other" category. With increase in the training set size, GPT2-intermediate and GPT2-medium are able to match/outperform GPT2-small on most evaluations. In particular for Train-L, GPT2-intermediate outperforms GPT2-small on all evaluations.

These results are along the expected lines of larger training sets alleviating the overfitting problem with larger models (Kaplan et al., 2020). Based on prior work and extrapolating from the current results, we expect GPT2-medium to outperform GPT2-intermediate for training sets sufficiently bigger than Train-L. Note that we stick with the default GPT2 configuration for all our experiments. Tuning the regularization hyperparameters such as dropout, can improve results for bigger models.

# D    ILLEGAL MOVE ANALYSIS

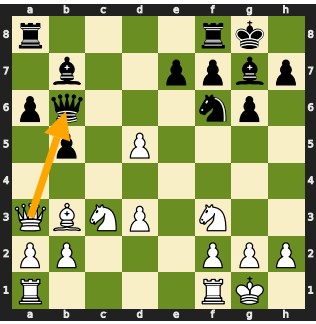

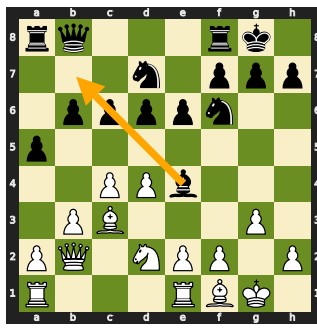

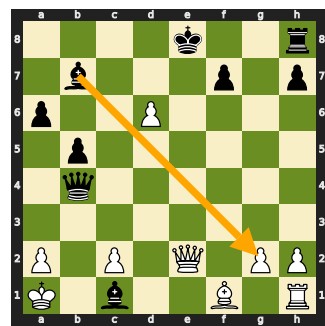

(a) *Syntax*: Predicted move is not possible for a queen.

(b) *Path Obstruction*: The pawn at `c6` is blocking the bishop.

(c) *Pseudo Legal*: The black king remains in check.

Figure 2: Visualization of the three categories of illegal moves.

Table 8: Error Analysis of Illegal Moves for models trained on Train-L

| Model | Ending Square (Long) | | | | | |
| --- | --- | --- | --- | --- | --- | --- |
| | Actual | | | Other | | |
| | Syntax | Path Obst. | Pseudo Leg. | Syntax | Path Obst. | Pseudo Leg. |
| UCI | 3 | 21 | 31 | 8 | 34 | 59 |
| UCI + RAP 5 | 1 | 18 | 25 | 6 | 26 | 51 |
| UCI + RAP 15 | 2 | 23 | 24 | 7 | 31 | 46 |
| UCI + Multi-view Training | 0 | 14 | 27 | 5 | 17 | 55 |
| Oracle Baseline | 0 | 15 | 11 | 2 | 31 | 27 |

Illegal moves can be exhaustively categorized into three types:

- *Syntax*: The predicted move can't be made by the piece type present at the starting square regardless of the board state. This error indicates that the model either fails at tracking the piece type present at the starting square or it lacks the spatial understanding of the board or both. These errors are rare; figure 2a presents one such case. In this particular case, many of the previous moves involved the `b6` square, which might have led to this error.

- *Path Obstruction*: The move can't be executed because there are other pieces present in the path. This error indicates the model's failure at tracking the board state. In figure 2b, the pawn at `c6` blocks the bishop's move from `e4` to `b7`.

- *Pseudo Legal*: This is a fairly tricky category where the move is illegal because the king ends up in check or remains in check if the predicted move is executed. We hypothesize that training explicitly on negative examples might be helpful/required in preventing these errors (Linzen et al., 2016; Noji & Takamura, 2020).

Table 8 presents results on analysis of illegal moves predicted by the model trained on the full training set Train-L for the ending square task for long histories. We observe that most of the gains for the oracle baseline are coming from the pseudo legal error category. The oracle baseline benefits from the input representation explicitly encoding the information that the king is in check. While the models are pretty closely matched on the syntax error category, in the path obstruction category the clear winner is the multi-view training setting, which suggests that the multi-view approach is better at modeling the spatial arrangement of the pieces.

# E  MISCELLANY

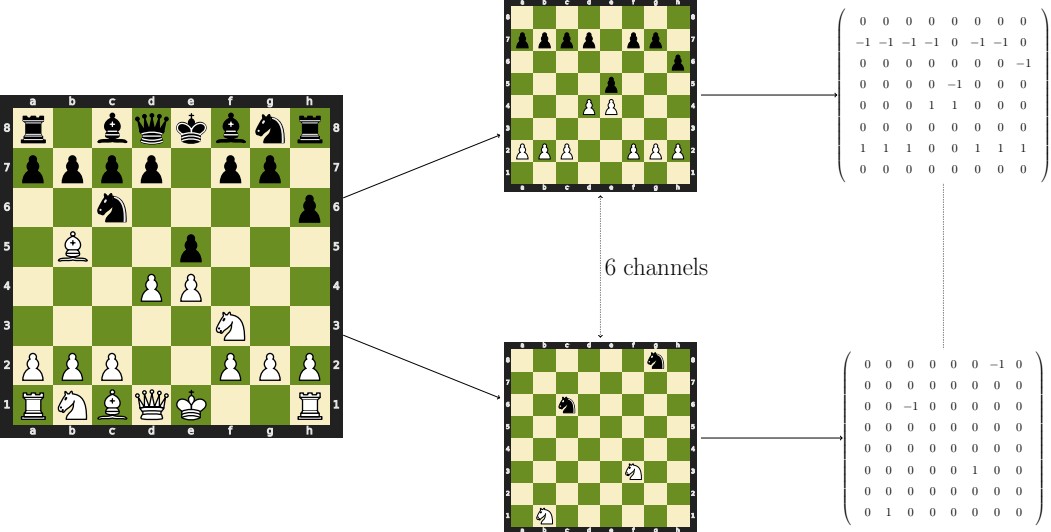

Figure 3: Part of the raw board representation used as input to the board state model.

Table 9: Statistics of language modeling data

| Split | # of games (in $10^3$) | Total # of moves (in $10^6$) | Avg. # of moves per game |
|---|---|---|---|
| Train-S | 15 | 1.1 | 73.6 |
| Train-M | 30 | 2.2 | 73.5 |
| Train-L | 100 | 7.3 | 73.5 |
| Dev | 15 | 1.2 | 78.5 |
| Test | 15 | 1.2 | 79.2 |

Table 10: Canonical Dev and Test perplexity. By canonical we mean that one move, say `f1b5`, counts as one token.

| Training Set | Model | Dev set | Test set |
|---|---|---|---|
| Train-S | UCI | 3.21 | 3.20 |
| | UCI + RAP 5 | 3.07 | 3.07 |
| | UCI + RAP 15 | 3.07 | 3.08 |
| | UCI + Multi-view Training | 3.22 | 3.22 |
| | Oracle Baseline | 2.71 | 2.71 |
| Train-M | UCI | 2.83 | 2.83 |
| | UCI + RAP 5 | 2.76 | 2.75 |
| | UCI + RAP 15 | 2.79 | 2.79 |
| | UCI + Multi-view Training | 2.80 | 2.80 |
| | Oracle Baseline | 2.46 | 2.46 |
| Train-L | UCI | 2.24 | 2.24 |
| | UCI + RAP 5 | 2.76 | 2.75 |
| | UCI + RAP 15 | 2.44 | 2.44 |
| | UCI + Multi-view Training | 2.29 | 2.29 |
| | Oracle Baseline | 2.13 | 2.13 |

## F   RESULTS FOR LSTM LANGUAGE MODELS

Table 11: Accuracies (%) for predicting ending squares for LSTM language models

| Training Set | Model | Ending Square (Short) | | | Ending Square (Long) | | |
| | | Actual | | Other | Actual | | Other |
| | LSTM-LM | ExM | LgM | LgM | ExM | LgM | LgM |
|---|---|---|---|---|---|---|---|
| Train-S | + UCI | 43.1 | 77.7 | 68.1 | 23.8 | 60.0 | 52.5 |
| | + UCI + RAP 5 | 42.0 | 77.1 | 67.1 | 23.5 | 59.8 | 53.2 |
| | + UCI + RAP 15 | 42.7 | 77.0 | 69.0 | 22.9 | 59.3 | 52.2 |
| Train-M | + UCI | 48.5 | 82.7 | 76.9 | 29.7 | 65.0 | 58.7 |
| | + UCI + RAP 5 | 50.1 | 83.8 | 77.3 | 27.5 | 64.0 | 59.0 |
| | + UCI + RAP 15 | 51.1 | 84.0 | 77.3 | 28.8 | 66.2 | 60.8 |
| Train-L | + UCI | 59.9 | 94.2 | 88.5 | 36.1 | 81.6 | 74.8 |
| | + UCI + RAP 5 | 59.4 | 93.5 | 87.7 | 38.1 | 83.3 | 75.7 |
| | + UCI + RAP 15 | 61.5 | 94.3 | 87.7 | 39.5 | 84.7 | 78.4 |

In this section we present results for LSTM language models (Hochreiter & Schmidhuber, 1997). In particular we use a LSTM model with 3 layers, 1024-dimensional hidden units, 768-dimensional input embedding size, and dropout of 0.1 applied to the output of the embedding layer and intermediate LSTM layers.[13] The motivation behind these experiments is to compare LSTMs and transformers in their ability to utilize the board state signal available via RAP during training. We focus on the RAP setting because with RAP there's a distribution shift between training and inference, and we want to compare how the two model classes handle this shift.

Table 11 presents results for LSTM language models, the corresponding numbers for transformers are presented in Table 4 in the main text. For Train-S, the vanilla UCI model outperforms both the RAP settings on all but a couple of Other LgM evaluations. For Train-M, the addition of RAP improves performance in all evaluations, except ExM accuracies for long histories. And finally for Train-L, both UCI + RAP 15 and UCI + RAP 5 outperform UCI over long histories. For short histories, UCI outperforms or matches both UCI + RAP 5 and UCI + RAP 15 on LgM evaluations, and only UCI + RAP 15 has an edge over UCI in ExM accuracy.

These results are in sharp contrast to the significant performance gains reported earlier for transformers with the addition of RAP, especially for smaller training sets. Thus transformers are better at utilizing RAP than LSTMs, and that transformers are more robust than LSTMs to RAP-like changes in input distribution. Our reasoning is that unlike LSTMs/RNNs, transformers have only a weak dependence on positions via position embeddings, and thus can better handle RAP-like changes where tokens can be randomly inserted/deleted throughout the length of the input sequence.

---

[13]Hyperparameters were chosen via grid search with dev perplexity as the evaluation metric

