# OpenReview forum: "Learning Chess Blindfolded"
_ICLR.cc/2021/Conference — Reject_

### Official Review · AnonReviewer2 · 2020-10-16
**Interesting testbed and surprising results!**

**Rating:** 7
**Confidence:** 4

**Review:**

Summary: This paper is an interesting exploratory study analyzing the ability of language models to track the state of a chessboard. The authors adopt a clever chess notation which allows them to probe the language model's state tracking ability by looking at its next word prediction (akin to probes in [1]). Quite remarkably, language models finetuned on chess data store a very accurate state representation, and predict legal moves over 90% of the times even without a visual representation of the board.

-----------------------------------

Strengths of the Paper:

1. A clever probing method to analyze language model's state tracking abilities.

2. Well-designed experiments and very interesting results in a mostly unstudied area.

3. Wel written paper with several baselines and ablations.

-----------------------------------

Weaknesses of the Paper / Possible additional analysis:

While the work in itself is very interesting and clean, I would have loved to see more analysis studying the model. This a very rich testbed where a lot of interesting experiments can be done! For instance,

(1) Does model performance / chess quality improve with larger language models like GPT2-md or GPT2-l?
(2) What are the kinds of errors these language models make (in terms of legal moves)? Do these errors disappear when you check top-k tokens?
(3) What are the kinds of moves the model is good at (when considering argmax predictions)? Is it learning any strategy at all? You can measure this quite well automatically using the chess engine scores which indicate who is winning, and comparing the change in scores when the actual move is played vs the language model's predicted move
(4) Finally can insights from (2) and (3) be transferred to other real-world applications? Is there a correlation between the probing literature on natural language processing tasks and the results you find?
(5) It will be cool to check other kinds of visual state fusion strategies like pseudo-self attention [2]

-----------------------------------

Other Feedback:

1. on the bottom of page 6, did you mean Table 2?
2. A couple of baselines will be useful in Table 2 and 3. The first could be upperbound EM baselines using engines like Stockfish or AlphaGo (since nearly everyone is worse at Chess than them, I expect the EM score to be lower than 100%). The second could be a random lowerbound to EM, where a random move from the set of LM is chosen. Finally it will be good to see EM performance of GPT-2 considering only the set of LM.
3. LM is an overloaded acronym which can cause confusion to the reader (language model vs legal move).

-----------------------------------

Overall Recommendation:

This is an exciting and rich testbed with a lot of interesting questions to answer. The authors have conducted well-thought experiments and reported interesting results. I'm leaning accept, but I encourage the authors to keep working on this setup (perhaps using some of the suggestions discussed above) and try to check if any of the insights here can be transferred to better understanding of language models on natural language.

-----------------------------------

References:

[1] - https://www.mitpressjournals.org/doi/pdfplus/10.1162/tacl_a_00115
[2] - https://arxiv.org/pdf/1908.06938.pdf

---

> ### Author Response · Authors · 2020-11-19
> **More Analysis Added**
>
> We thank the reviewer for their positive and helpful comments.
>
> > Does model performance / chess quality improve with larger language models like GPT2-md or GPT2-l?
>
> We have added results with larger models (upto GPT2-medium) in Appendix C.2. For bigger training sets there are minor gains with larger models.
>
> > What are the kinds of errors these language models make (in terms of legal moves)? Do these errors disappear when you check top-k tokens?
>
> A detailed analysis of illegal moves for the end square prediction is done in Appendix D. We find two prominent error categories: (a) piece(s) obstructing the predicted move, and (b) the current player’s king is in check or remains in check after the predicted move is executed.
>
> > What are the kinds of moves the model is good at (when considering argmax predictions)? Is it learning any strategy at all? You can measure this quite well automatically using the chess engine scores which indicate who is winning, and comparing the change in scores when the actual move is played vs the language model's predicted move
>
> Chess strategy evaluation would be an interesting analysis. However, right now the language model is trained to predict both the winning and losing moves. We would’ve trained the language model in a different way if the focus was trying to train a better chess player. Specifically, adding a prefix bit to indicate the winning player i.e. black or white (and conditioning on that during gameplay) or suppressing the language model loss corresponding to the losing player. This is outside the scope of our current work but an interesting future direction.
>
> > Finally can insights from (2) and (3) be transferred to other real-world applications? Is there a correlation between the probing literature on natural language processing tasks and the results you find?
>
> We plan to apply findings from the chess testbed to natural language applications. The RAP experiments have demonstrated that incorporating parts of the world state as tokens in the original sequence is a very effective strategy (similar in spirit to pseudo self attention). In future work, we want to incorporate coreference chains in text via special [ENTITY_i] tokens, where i is the cluster ID, and finetune pre-trained encoders. Similar to RAP, these tokens provide a slice of the world state and can be used as prompts to test the model’s understanding.
>
>  >  It will be cool to check other kinds of visual state fusion strategies like pseudo-self attention.
>
> Pseudo-self attention indeed offers an interesting way of incorporating visual modality.  Due to limited time we won’t be able to test it out in the rebuttal phase but thanks for the suggestion.
>
> > on the bottom of page 6, did you mean Table 2?
>
> Yes, thanks for pointing that out.
>
> > A couple of baselines will be useful in Table 2 and 3. The first could be upperbound EM baselines using engines like Stockfish or AlphaGo (since nearly everyone is worse at Chess than them, I expect the EM score to be lower than 100%). The second could be a random lowerbound to EM, where a random move from the set of LM is chosen. Finally it will be good to see EM performance of GPT-2 considering only the set of LM.
>
> We have added the random legal move baseline to the table. Selecting the top legal move improved the performance on exact end move prediction by UCI in the Train-L setting by 1% absolute in short histories, and 2.7% absolute in long histories. We will add that and the chess engine baseline in the final version.
>
> > LM is an overloaded acronym which can cause confusion to the reader (language model vs legal move).
>
> That's a great suggestion. We now refer to legal move as LgM, and exact move as ExM.
>
> We have added the suggested references.
> Let us know if you have any additional questions or concerns.

---

> > ### Comment · AnonReviewer2 · 2020-11-20
> > **Thanks for the replies!**
> >
> > Thank you for the detailed replies and adding more baselines. I will stick to my accept score of 7 since I really enjoyed reading the work and thought it had thorough experiments! I won't raise my score further since I sort of agree with other reviewers (partly voiced in my weakness #4) that the work is a bit narrow and does not have too many insights which can be applied to real-world applications. But I appreciate the fact that you have proposed future work in this direction.

---

> > > ### Author Response · Authors · 2020-11-23
> > > **Follow up**
> > >
> > > We appreciate your positive comments.
> > >
> > > To reiterate the wider implications of this work, our results shed light on the following interesting properties of transformers: (a) they are robust to RAP-like changes in input distribution, and (b) they require access to long context (Appendix C.1), and large training sets. Future work can use the first finding to introduce the world state, or more specifically the output of linguistic analyzers such as coreference, via RAP-like tokens during pre-training and fine-tuning of transformers. RAP-like tokens can also be used for debugging/diagnosing the model’s understanding, similar to the starting square prediction tasks. The second finding can be another motivation to search for new architectures that are adept at understanding long text with access to limited history (Rae et al., 2020), and that require small training sets. The framework we have introduced allows for probing and understanding new architectures that address these challenges.
> > >
> > > Additionally, we confirm in Appendix F that transformers are more robust than LSTMs to changes in input distribution due to RAP. This is because unlike LSTMs/RNNs, transformers have only a weak dependence on positions via position embeddings.  This makes us optimistic about the proposed future work direction.

---

### Official Review · AnonReviewer4 · 2020-10-28
**Interesting new benchmark that would benefit from more connections to the literature.**

**Rating:** 5
**Confidence:** 3

**Review:**


### Topic
This paper explores learning chess from raw notation as a benchmark for the ability of language models to track world state. Chess is an interesting benchmark, as a set of moves can be unambiguously linked to a world state, there are large amounts of data available and the model can easily be probed for its board tracking abilities. The contributions of this paper are twofold: (i) introducing blindfolded chess as a benchmark for grounded language learning and world state tracking, as well as a suite of probing tasks to evaluate models; (ii) empirical evidence that transformer language models can learn both the rules of the game and to track board state.

### Pros:
-	Blindfolded chess is an interesting benchmark for grounded language learning and as a testbed for models to track world state. It is unambiguous, data-rich, has a limited vocabulary and models trained on it can easily be probed. This adds to prior papers on learning chess with transformers that have mainly focused on the performance of such models.
-	The use of SAN + RAP trick is an interesting one to be able to probe the current location of pieces.
-	The oracle model is interesting, as it demonstrates the gap between a model that must track world state and one that has access to it. The multi-view model is also interesting as an example of how additional supervision may help the model.
-	The analysis across different data sizes and game length is interesting.

## Cons:
-	The paper feels rushed at times: for instance, there are no references in text to Appendix D, despite it being an interesting demonstration of the analysis that can be done in this environment.
-	The comparison to related work on world state tracking and grounded language learning could be substantially improved. The authors only very briefly mention previous work: the papers on the TextWorld environment (Cote et al, 2018), seem relevant, although the framework is that of interactive environments and RL. Other papers on grounded language learning also seem relevant, such as Alexander G. Ororbia II et al, 2019. More so than the lack of references, an issue with this paper is that few connections are made to wider research issues in grounded language learning / implicit world state tracking. Instead, the text tends to simply state experimental results. This makes the contribution of this paper very narrow.
-	It is not clear why the multi-view trained models underperform on the low data settings compared to the models without the extra supervision. If there is more explicit supervision of the model, should it not be better at tracking world state?
-	It would be good to make clear if this dataset and probing tasks will be released, and the code made available. This would be a welcome step in standardizing this new domain, as prior approaches have all used different settings/datasets/notations/evaluations.
-	Despite this being a GPT-small architecture and the authors using a small subset of the available training data, the accuracies are high for legal move predictions on Train L. Those are the ones that are directly related to tracking the world state and knowing which piece movements are allowed. This limits the use of this task as a future benchmark for world state tracking in the data-rich setting. It might be possible to make the task more challenging by asking the language model to predict the entire board state (deconvolution) or by focusing on hard subsets of the task (e.g: pseudo-legal infractions on long histories).
-	Many challenges of natural language are not present in this environment, such as coreference, limiting the applicability of results on this task to e.g bAbl. The use of a composite vocabulary (e.g: “e4” instead of “e”, “4”) also limits the compositional challenges of the dataset. However, it is true that using a vocabulary where “e4” is two tokens would make probing more difficult.

Minor comments:
-	Incomplete board state: The board state as represented is actually not complete, as it ignores whether a pawn can be taken en passant ( https://en.wikipedia.org/wiki/En_passant ). En passant is rare enough that this might not matter, but it does mean the Oracle/Multi view models do not perfectly capture the state. Note that this seems to also be an issue in Oshri and Khandwala (2015) from whom this representation is derived.
-	The ending square task does not allow us to probe whether the model captures the full range of possible moves for a piece, or whether it selects a subset of those. E.g: the model might reach 100% accuracy on this task without ever moving a rock/bishop/queen by more than one square (less likely that there are pieces in between).
-	It is not clear what you do with games of length 100-150. These seem included in the training set but excluded from the probes. Are those included in the perplexity results of Table 6.
-	Typos:
o	Table 4 breakdown in appendix B

### Recommendation:

I lean reject for this paper.
The core idea is interesting, but the paper fails to make connections to wider issues in world state tracking and grounded language learning, making it overly narrow. Both the missing references and the missing links to wider concepts in this litterature are a symptom of that. With a few caveats, the experiments are sound, but the analysis could be improved to go further than simply stating the results. The overall wording and presentation of the paper must also be improved.

Questions:
- Are you planning to release the code and data to facilitate work on this topic?



## Author response update

In light of the author response, I have decided to increase the score to 5. I have also decreased my confidence to 3.
The main reasons for this score increase are the release of code and data as well as thoughtful clarifications on the experimental setup. This is good experimental work. I also think Appendix C.1 is a good first step towards drawing wider scientific conclusions from this work.
 The main reason not to increase the score further is that I believe the contribution still is quite narrow.

I chose to decrease confidence in my evaluation since it is now based more on the narrowness of the contribution, which is harder to assess, than on the experimental validity of this work.

---

> ### Author Response · Authors · 2020-11-19
> **Added connections to literature**
>
> We thank the reviewer for their detailed and constructive feedback. Below we have tried to address the reviewer’s concerns.
>
> > No references in text to Appendix D, despite it being an interesting demonstration of the analysis that can be done in this environment.
>
> We have added more details to the illegal move analysis section and also added references to the same in the main text (Appendix D).
>
> > The comparison to related work on world state tracking and grounded language learning could be substantially improved.
>
> We appreciate the pointers to related work and agree that we could be more expansive in describing connections with and differences from this prior work.  We have accordingly significantly expanded the related work section. With regards to the specific references suggested by the reviewer, we quote the following text from the related work section.
>
> TextWorld-style environments resemble ours in that the true world state is, by construction, available, models trained for a TextWorld environment differ in:
> 1. their objective (reward maximization vs. maximizing the probability of the next observation),
> 2. in how they are ultimately evaluated (final reward vs. world state tracking), and
> 3. in whether we can directly probe the model’s knowledge of the entire state.
>
> The world models of Ha & Schmidhuber (2018) also maximize the probability of the next observation, but differ along the other two dimensions. Similarly the work by Hermann et al. (2017) and Hill et al. (2017) on developing and using 3D world simulations for learning grounded language has only partial overlap with the objective function, and differ along the other two aspects.
>
> Our work is related to work on grounding in that we are interested in comparing model performance when it does not have access to grounding information to when it does (Bruni et al., 2014; Kiros et al., 2014; Ororbia et al., 2019). However, unlike the work of Ororbia et al. (2019), for instance, the goal is not to improve the performance of language models using access to more of the world state, but to assess how much of this state has been recovered by the model from just learning the language modeling task.
>
> > It is not clear why the multi-view trained models underperform on the low data settings compared to the models without the extra supervision.
>
> Regarding multi-view trained models, our guess is that the board state model may require more data than it is being given in Train-S to train well. Without enough data, a poorly learned representation for board state can adversely interact with the language model. Pretraining the board state model might be a useful strategy. However, in an earlier experiment using a fully-connected network for board state, pretraining the network didn’t help.
>
> > It would be good to make clear if this dataset and probing tasks will be released, and the code made available.
>
> The entire setup is available at [this URL](https://anonymous.4open.science/r/f0c718e1-16af-4f1a-a129-d4ace9ac6820/). We chose a publicly available chess database to keep the setup reproducible. We also plan to release pretrained models via the huggingface model hub.
>
> > Despite this being a GPT-small architecture .... make the task more challenging ...
>
> The tasks can be made more challenging in various ways, including the suggestion of the reviewer to predict the whole board state. Some other possible ways are: (a) limiting the attention window of the transformer as chess is almost Markovian (results in Appendix C.1) (b) evaluating on even longer game histories, and (c) focusing on Train-S which we refer to as small but still has more than a million moves.
>
> > Many challenges of natural language are not present in this environment.
>
> We agree that chess notation is incredibly simple compared to natural language. The goal of our work was to start with a simple setting to determine how well transformer language models are tracking the world state, and how this state tracking capability can be improved with access to the world state during training. We found RAP to be an effective and easy way of adding slices of the world state to the text sequence, which improves performance and can later be used for diagnosing/probing the learned models. In our future work, we plan to explore adding entity-related information, such as coreference chains, via RAP-like tokens during pre-training of natural language models.
>
> > Incomplete board state: The board state as represented is actually not complete, as it ignores whether a pawn can be taken en passant
>
> The board state representation is indeed incomplete. Information such as the possibility of en passant, counter for threefold repetition, counter for fifty move rule, etc., are missing from the board state. Since this information is needed only in rare cases, we didn’t use it as part of the board state. We have mentioned this in a footnote in the revision.
>
> **Response continued in later comment (limit on characters)**

---

> > ### Author Response · Authors · 2020-11-19
> > **(Response continued) Trivial Emulation vs Actual Understanding**
> >
> > > The ending square task does not allow us to probe whether the model captures the full range of possible moves for a piece, or whether it selects a subset of those. E.g: the model might reach 100% accuracy on this task without ever moving a rock/bishop/queen by more than one square (less likely that there are pieces in between).
> >
> > This is a very valid concern. However, the models do achieve reasonably high accuracy on the exact move task which makes us think that it's unlikely that the model is just making trivial predictions.
> > We also conducted a preliminary analysis to verify this. The analysis focused on the distance between the starting and ending square in "king-moves" ([Python-chess implementation](https://python-chess.readthedocs.io/en/latest/core.html#chess.square_distance)). We only included moves made by rook, bishop, and queen in this analysis because both king and knight make moves constant in king-move distance (pawns were already excluded due to their simple dynamics).   We first present statistics for the ground truth:
> > * Actual Ending Square (Short); Filtered prompts (rook, bishop, queen) = 658; Average king-move distance between starting square and ending square = 2.0; Percentage of prompts with ending square at king-move 1 distance =  48.9
> > * Actual Ending Square (Long); Filtered prompts (rook, bishop, queen) = 697; Average king-move distance between starting square and ending square = 2.2; Percentage of prompts with ending square at king-move 1 distance =  42.9
> >
> > Next, we present statistics for the UCI model trained on Train-L:
> > * Actual Ending Square (Short); Filtered prompts (rook, bishop, queen) = 658; Average king-move distance between starting square and predicted ending square = 2.0; Percentage of prompts with ending square at king-move 1 distance =  50.8
> > * Actual Ending Square (Long); Filtered prompts (rook, bishop, queen) = 697; Average king-move distance between starting square and predicted ending square = 2.1; Percentage of prompts with ending square at king-move 1 distance =  43.6
> >
> > Filtering the predicted moves by whether they were correct or not didn't change the stats by much.
> >
> > To summarize:
> > * The average distance covered in ground truth moves made by rook/bishop/queen for short game histories is about 2 (upper-bound of 7)
> > * In long histories the average distance covered is slightly higher. This is quite possible because as the game proceeds, there are lesser pieces on the board and thus lesser obstructions.
> > * For both histories, more than 40% of moves are within 1 king-move
> > * The predicted moves are slightly closer on average in comparison to ground-truth moves.
> > * More often than not, the predicted ending squares are more than 1 king-move distance away.
> >
> > The broader implications of LM merely emulating chess without understanding chess still needs more investigation.
> > In particular, we plan to investigate the quality of top K predictions of the model where K could either be the number of legal moves possible in the position or all the moves above a chosen probability threshold.
> >
> > > It is not clear what you do with games of length 100-150. These seem included in the training set but excluded from the probes. Are those included in the perplexity results of Table 6.
> >
> > The games of length 100-150 are indeed used for perplexity calculation. These games can also be used in the future to develop harder evaluations.
> >
> > We hope this clarifies the concerns raised above. Let us know if you have any additional questions or concerns.

---

> > > ### Comment · AnonReviewer4 · 2020-11-20
> > > **Updated score to 5 in light of code release + experimental clarifications, decreased confidence to 3 since score is mostly a function of the narrowness of the contribution, which is more subjective.**
> > >
> > > In light of the author response, I have decided to increase the score to 5. I have also decreased my confidence to 3.
> > > The main reasons for this score increase are the release of code and data as well as thoughtful clarifications on the experimental setup. This is good experimental work. I also think Appendix C.1 is a good first step towards drawing wider scientific conclusions from this work.
> > >  The main reason not to increase the score further is that I believe the contribution still is quite narrow.
> > >
> > > I chose to decrease confidence in my evaluation since it is now based more on the narrowness of the contribution, which is harder to assess, than on the experimental validity of this work.

---

> > > > ### Author Response · Authors · 2020-11-23
> > > > **Follow up**
> > > >
> > > > We thank the reviewer for their increase in score.
> > > >
> > > > Regarding the wider implications of this work, we want to say that our results shed light on the following interesting properties of transformers: (a) they are robust to RAP-like changes in input distribution, and (b) they require access to long context (Appendix C.1), and large training sets. Future work can use the first finding to introduce the world state, or more specifically the output of linguistic analyzers such as coreference, via RAP-like tokens during pre-training and fine-tuning of transformers. RAP-like tokens can also be used for debugging/diagnosing the model’s understanding, similar to the starting square prediction tasks. The second finding can be another motivation to search for new architectures that are adept at understanding long text with access to limited history (Rae et al., 2020), and that require small training sets. The framework we have introduced allows for probing and understanding new architectures that address these challenges.
> > > >
> > > > Additionally, we confirm in Appendix F that transformers are more robust than LSTMs to changes in input distribution due to RAP. This is because unlike LSTMs/RNNs, transformers have only a weak dependence on positions via position embeddings.

---

### Official Review · AnonReviewer1 · 2020-10-29
**Review 1**

**Rating:** 5
**Confidence:** 4

**Review:**

Summary: This paper explores the abilities of transformer-based models to do grounded state tracking via chess. They train a GPT-2 on chess games, showing that it can learn the rules of the game by being able to state-track and predict valid next actions.

Pros:
1. A very thorough set of experiments are given to explore how transformers can be used to track states in a game such as chess and results show that with enough data, transformers can predict the locations of pieces across a decent amount of history as well as predict legal actions.
2. The paper is well-written in terms of general writing clarity and I was able to follow *what* was happening throughout.

Cons:
1. I am a bit lost as to the motivation and positioning behind the paper, i.e. I was unsure as to *why* things were happening as they were. The authors say that they are using transformers to see how transformers can learn grounded language when world states are available but I do not see this work positioned with respect to other work on grounded language nor work on state tracking generally found in model-based RL.
(i) An example of the former for grounded language learning (given that they cite Bender and Koeller) would be instances such as vision and language navigation (Anderson et al. https://openaccess.thecvf.com/content_cvpr_2018/papers/Anderson_Vision-and-Language_Navigation_Interpreting_CVPR_2018_paper.pdf), the Nethack Learning Environment for game grounding (Kuttler et al. https://arxiv.org/abs/2006.13760), or text games (Cote et al. https://arxiv.org/abs/1806.11532 and Hausknecht et al. https://arxiv.org/abs/1909.05398). I am not sure how state tracking in chess implies that transformers can do grounded language learning.
(i) In terms of just the state tracking parts, there has already been much work in agents that learn the rules of the world as they play through them. This can happen with World Models (Ha and Schmidthuber https://arxiv.org/abs/1803.10122) or in cases like Alpha-Zero (Silver et al. open access version https://kstatic.googleusercontent.com/files/2f51b2a749a284c2e2dfa13911da965f4855092a179469aedd15fbe4efe8f8cbf9c515ef83ac03a6515fa990e6f85fd827dcd477845e806f23a17845072dc7bd) which learn the rules of Chess from scratch via self play. How does using a transformer compare to these works?
2. Given the above and the fact that all of the pieces of the methodology of this work are taken from others (the architecture, training, state representation, etc.) - the main contribution is the experimental design and the results themselves. In this case, I would have liked to see more analysis regarding exactly what properties of the transformer they think is responsible for helping the model to learn and also a potential qualitative analysis of what the failure cases are.

Overall, the paper has some interesting ideas, experiments, and results but is not connected to the motivation/is not positioned well with respect to closely related work.

Post author response:
See comment below for further score justification.

---

> ### Author Response · Authors · 2020-11-19
> **Clarifications on positioning of the work**
>
> We appreciate the pointers to related work and agree that we could be more expansive in describing connections with and differences from this prior work.  We have accordingly added this discussion to the related work section.
>
> In relating our work to this other work we emphasize that a major goal of this paper is to determine how well models trained just on symbolic inputs (with a language modeling objective) are able to track the world state underlying these symbolic inputs. Certain probing papers also have a similar goal, but Chess/UCI is unique in that we can precisely automate probing tasks and can evaluate them at a fine-grained level, and this probing requires no extra machinery that also may give less direct answers.
>
> While TextWorld-style environments resemble ours in that the true world state is, by construction, available, models trained for a TextWorld environment differ in:
> 1. their objective (reward maximization vs. maximizing the probability of the next observation),
> 2. how they are ultimately evaluated (final reward vs. world state tracking), and
> 3. whether we can directly probe the model’s knowledge of the entire state.
>
> The world models of Ha & Schmidhuber (2018) also maximize the probability of the next observation, but differ along the other two dimensions. Similarly the work by Hermann et al. (2017) and Hill et al. (2017) on developing and using 3D world simulations for learning grounded language has only partial overlap with the objective function, and differ along the other two aspects.
>
> Our work is related to work on grounding in that we are interested in comparing model performance when it does not have access to grounding information to when it does (Bruni et al., 2014; Kiros et al., 2014; Ororbia et al., 2019). However, unlike the work of Ororbia et al. (2019), for instance, the goal is not to improve the performance of language models using access to more of the world state, but to assess how much of this state has been recovered by the model from just learning the language modeling task.
>
> With regards to AlphaZero, our setup, goals, and training objectives are very different.
> * AlphaZero has access to the board state and rules of chess. AlphaZero starts with random play governed by rules of chess, and learns from just self-play (Without the knowledge of rules of chess and access to actual games it’s impossible to learn anything meaningful). On the other hand, our models don’t have access to the board state during inference (except for the theoretical oracle baseline) and don’t know the rules of chess.
> * A major goal of this paper is to determine how well models trained just on symbolic inputs (with a language modeling objective) are able to track the world state underlying these symbolic inputs. This is very different from AlphaZero’s goal to demonstrate the potential of self-play in closed domains.
> * AlphaZero’s training objective is to predict the next move that maximizes the reward, where reward is winning the game, while our training objective is to maximize the probability of the next observation, regardless of the quality of the play.
>
> We hope this clarifies the positioning of our work w.r.t. prior literature.
>
> > I would have liked to see more analysis regarding exactly what properties of the transformer they think is responsible for helping the model to learn and also a potential qualitative analysis of what the failure cases are.
>
> * We present additional results with variations in the basic transformer architecture (GPT2-small) in Appendix C. In Appendix C.1, we compare the effect of limiting access to previous tokens. Even though chess is Markovian, we find that there’s a significant performance drop with limited history. This demonstrates the dependence of the transformer LM's performance on access to unrestricted history, and its limited ability to learn a compressed state representation for the relatively simple domain of chess.
> * Appendix D has a detailed illegal move analysis, and we have added references to the same in the main text.

---

> > ### Comment · AnonReviewer1 · 2020-11-23
> > **Response to Rebuttal**
> >
> > I appreciate the authors efforts to clarify my questions and revise their manuscript.
> >
> > I am satisfied with the answers given differentiating this work from Alpha Zero as well as the additional experiments performed. I would contend though that the differences with the other environments I have provided in point 1 of my initial review are not sufficient. The three dimensions given with respect to differences with TextWorld (and hold for the other envs too) are not entirely accurate. There is nothing in the framework itself that focuses on reward maximization instead of next state probability - its the same as having a chess simulator where you can either focus on predicting the next state or just have an external reward the indicates whether or not you've won the game. It is possible to generate oracle traces, etc. equivalents to this chess dataset in most of these frameworks.  Overall that is to say, chess can also be framed in exactly those three terms and given that these are frameworks and not agents, you cannot say that these three dimensions hold.
> >
> > This being said, in appreciation of the author's efforts for the other clarifications - I will increase my score to a 5.

---

> > > ### Author Response · Authors · 2020-11-24
> > > **Comparison with other frameworks and contributions of the work**
> > >
> > > We thank the reviewer for their appreciation of our effort and the increase in score.
> > > Our claims about the frameworks were based on the intended use case. While it may be possible to use frameworks like TextWorld to just predict observations, this is not what the original work or the follow-up work has done. Moreover, it is not clear to us that TextWorld-like environments allow for probing the model in the same way.
> > >
> > > Our work demonstrates that in chess we have a tailor-made domain where models can be trained with the language modeling loss, and evaluated on world state tracking via simple prompts. We are not aware of any other framework that readily allows for this. Moreover, our contributions are not just limited to proposing this new framework. The results of using the proposed framework provide insights into how effective transformers are at tracking the world state, their robustness to input perturbations, and, perhaps most importantly, the remaining challenges facing transformer LMs even in this relatively simple domain.

---

### Official Review · AnonReviewer3 · 2020-10-29
**An interesting application of transformers to Chess playing**

**Rating:** 7
**Confidence:** 4

**Review:**

This paper considers an intriguing problem: can language models, when trained on
purely textual representations of Chess games, learn the underlying dynamics of
the game? The authors argue that this could be a preliminary step towards
tackling the symbol grounding critique of methods like transformers. When
transformers are utilized in natural language settings, it's challenging to
determine whether the models are operating at a pure syntactic level, or whether
there is some rudimentary level of "understanding", given that the models are
only exposed to text. In contrast, in Chess, one can train in a purely textual
fashion (or with some limited symbol grounding information), and probe how well
the result models the state of the underlying Chess position.

The authors use the GPT2-small based transformer architecture and train from scratch on a dataset of
high-quality Chess games between humans, represented in UCI notation -- a
textual listing of the moves made by each player. In some experiments, this
purely textual input is supplemented with explicit board state information as
well, to test the impact of this additional signal. The authors evaluate the
system on two types of inference tasks: a trained model's ability to
successfully locate a piece on the board, and the model's ability to determine
where to move a chosen piece to next. In each case, there are two
evaluation metrics: an "exactness" metric (i.e., whether the model picked the
same piece/move as the human did in the corresponding game) and a "legality"
metric (i.e., whether the model picked a permissible move). The former metric is
more stringent, as it's also measuring the strategic awareness of the model. The
authors demonstrate through their experiments that transformers are successful
at both inference tasks with very high accuracy, particularly when evaluated
using the legality metric, using shorter sequences of moves, and larger
datasets.

Strengths of the paper:
  + This is a creative application of transformers to a non-traditional textual
    inference task. It may inspire others to devise other interesting applications.
    The results are intriguing and expand our understanding of what may be
    possible with transformer architectures.
  + The authors' approach is a fairly straightforward application of off-the-
    shelf techniques -- and I mean that in a good way. There are no unnecessary
    complications or ad hoc additions to the system design.
  + The paper is very clearly written, well-organized, and easy to follow.

Areas for improvement/questions for the authors:
  - I'm a little confused as to why performance peaks with p=0.05/0.15 in the
    UCI+RAP models. If my understanding is correct, then RAP with
    p=1.0 would include piece annotations with *every* move during training
    time and at inference. So shouldn't this make it easier to track pieces (as
    the authors themselves note in 4.1)? The reason for this is given as greater
    "mismatch between training and inference" -- I'm not sure what this means.
  - On a related note, I'm also confused by the performance of the Oracle Baseline
    in Table 3. In some instances, this is outperformed by trained models with
    *less* information. But my understanding was that this oracle would serve
    as an upper bound on model performance. So how are we outperforming the
    upper bound?

On balance, I think the strengths of the paper outweigh my concerns, and I
recommend ACCEPTANCE.

Couple of minor issues:
  - At the bottom of Page 6, the text references results in Table 3, which
    should be Table 2.
  - Section C in the Appendix is currently empty.

---

> ### Author Response · Authors · 2020-11-19
> **Clarification on RAP and Oracle Baseline**
>
> We thank the reviewer for their helpful and positive feedback. Below we answer the clarification questions asked by the reviewer.
>
> > I'm a little confused as to why performance peaks with p=0.05/0.15 in the UCI+RAP models. If my understanding is correct, then RAP with p=1.0 would include piece annotations with every move during training time and at inference. So shouldn't this make it easier to track pieces (as the authors themselves note in 4.1)? The reason for this is given as greater "mismatch between training and inference" -- I'm not sure what this means.
>
> In the RAP setting, the **piece types are only added during training** and not during inference, except in the prompt for the starting square prediction task. To clarify this we have added Table 1 with tokenized sequences. The motivation for including RAP in training is twofold:
> * It allows us to probe at test time, where the model thinks each piece is, by simply appending the piece type of interest to any game history's prefix.
> * We can use the available world state during training but the model doesn’t require it during inference.
>
> > On a related note, I'm also confused by the performance of the Oracle Baseline in Table 3. In some instances, this is outperformed by trained models with less information. But my understanding was that this oracle would serve as an upper bound on model performance. So how are we outperforming the upper bound?
>
> Regarding the performance of the oracle baseline:
> * We have made a minor change to the "oracle" baseline which has improved its performance, though it is still not the top performer in two evaluations with Train-L. In the earlier version, we used a limited attention window since the language model doesn’t have to track the board state as it’s already provided. There were empirical reasons as well:  The model with the limited attention window converged faster and hence did better than the model with access to the full history in the earlier stages of training. But we later found that the model with limited attention converged to an inferior perplexity than the one with access to the full attention history. This might be because access to all the previous tokens can better reveal insights about the players involved, and we do see improvements in the Exact move accuracies. We have updated the oracle model’s description and updated the numbers in the latest version.
> * The "oracle" baseline is intended to serve as an **approximate upper bound** where the model has access to more of the world state. It’s still a language model and can still make mistakes. The term "oracle" is probably a misnomer in our context given its typical use in the ML literature. We plan to change this in the final version but are keeping it for now to avoid too many changes. A more strict upper bound might use the constraints of chess to restrict prediction to legal moves and focus on just predicting the exact moves (suggested by Reviewer 2). That being said, the "oracle" model with access to board state doing slightly worse than some of the baselines is still intriguing. Our hypothesis is that the models are currently being trained and tested on both the long and short histories, almost like a multi-task setup. Different models can weigh these “different” tasks differently. Given the evaluation results, the oracle model may be prioritizing the long history tasks over the short history tasks.
>
> > Couple of minor issues:
> At the bottom of Page 6, the text references results in Table 3, which should be Table 2.
> Section C in the Appendix is currently empty.
>
> Thanks for pointing these out. We have fixed them in the revised version.

---

> > ### Author Response · Authors · 2020-11-23
> > **Additional comments on RAP**
> >
> > We wanted  to add a couple of more findings on RAP:
> > * In RAP, piece types are not added during inference. Though in earlier experiments, which we didn't report, the RAP models used piece types during inference and it certainly helped the model (as suggested by the reviewer). The reasons for not using piece types during inference have already been explained in the previous reply. Note that the oracle model has access to all the piece types during training and inference.
> > *  Additionally, we confirm in Appendix F that transformers are more robust than LSTMs to changes in input distribution due to RAP. This is because unlike LSTMs/RNNs, transformers have only a weak dependence on positions via position embedding.
> >
> > Hope this helps. Let us know if any further clarifications are needed.

---

### Author Response · Authors · 2020-11-19
**Paper Revision Overview**

We thank all the reviewers for their valuable feedback which we have tried to incorporate in the current revision. This comment is intended to give a consolidated view of all the changes made:
* We have substantially revised the related section work based on Reviewer 1 and 4's feedback.  We have tried to specify more clearly the contributions of our work in the revised introduction section.
* "Legal moves" are now denoted as LgM to avoid overloading of LM (Reviewer 2’s suggestion), and "Exact moves" use the ExM acronym rather than EM.
* A detailed analysis of illegal moves in Appendix D.
* Added a random legal move baseline for exact move evaluation as suggested by Reviewer 2.
* The entire experimental setup is available at [this URL](https://anonymous.4open.science/r/f0c718e1-16af-4f1a-a129-d4ace9ac6820/)
* Variations over the base transformer architecture (GPT2-small) are explored in Appendix C. Specifically, we report results with limited attention window + bigger models. We find that the success of the transformer model in our setting relies on access to the whole history, and the model suffers performance drop with limited attention window. This suggests that the model is not able to learn a compressed state representation even though chess is markovian and the board state can be described in less than 1K bits.
* We have made a minor change to the "oracle" baseline which has improved its performance. In the earlier version, we used a limited attention window since the language model doesn’t have to track the board state as it’s already provided. There were empirical reasons as well, the model with limited attention window converged faster and hence did better than the model with access to the full history in the earlier stages of training. But we later found that the model with limited attention converged to an inferior perplexity than the one with access to the full attention history. We have updated the oracle model’s description and the corresponding numbers. (More details in response to Reviewer 3)
* We plan to publicly release the trained models via the [Hugging Face modelhub](https://huggingface.co/models)

Finally, we want to reiterate the contribution of our work (quoted from revised Introduction):
* Propose chess as a testbed for evaluating world state tracking capabilities of language models.
* Show that by selecting (and tweaking) the appropriate chess notation, we can probe the language model for aspects of the world state using simple prompts (Section 3).
* Propose a suite of probing tasks to evaluate language models for chess on world state tracking (Section 5.3). These probing tasks go beyond simple exact match, and use a more fine-grained evaluation, and allow for automated error analysis (Appendix D).
* Show that given enough training data, transformer language models can learn to track piece locations and predict legal moves at high levels of accuracy.
* Evaluate the effect of grounding by training and evaluating a spectrum of transformer language models with varying degrees of access to the world state. We find that grounding helps in challenging settings of our proposed probing tasks.
* Provide insights on transformer language models such as their robustness to incorporating the world state in various ways, their dependence on access to long context, etc.

---

### Decision · Program_Chairs · 2021-01-07
**Final Decision**

**Decision:**

Reject

**Comment:**

I thank the authors for their submission and very active participation in the author response period. World state tracking is an important problem that encompasses existing problems like coreference resolution. I agree with R2 and R3 that proposing a novel environment in which we can investigate to what extend Transformers can tackle world state tracking should be interesting to the community. The majority of the reviewers agree that this paper presents an interesting benchmark [R2,R3,R4] with good thorough experimental work [R1,R2,R4]. However, R1 is confused about the positioning of the work and R4 finds the work narrow. R2, despite positive review, agrees with this assessment. I agree with this assessment as well and, after discussion with the program chairs, came to the decision that this paper is not ready for publication in its current state. I strongly encourage the authors to incorporate R1's and R4's feedback, in particular with respect to positioning this environment in comparison to TextWorld, and resubmit to the next venue.